# Multiparametric Dynamic Ultrasound Approach for Differential Diagnosis of Primary Liver Tumors

**DOI:** 10.3390/ijms24108548

**Published:** 2023-05-10

**Authors:** Maria Elena Ainora, Lucia Cerrito, Antonio Liguori, Irene Mignini, Angela De Luca, Linda Galasso, Matteo Garcovich, Laura Riccardi, Francesca Ponziani, Francesco Santopaolo, Maurizio Pompili, Antonio Gasbarrini, Maria Assunta Zocco

**Affiliations:** 1CEMAD Digestive Disease Center, Fondazione PoliclinicoUniversitario “A. Gemelli” IRCCS, Catholic University of Rome (Italy), 00168 Rome, Italy; 2Internal Medicine, University Hospital, 70100 Bari, Italy

**Keywords:** multiparametric ultrasound, dynamic contrast enhanced ultrasound, HCC, ICC, elastography

## Abstract

A correct differentiation between hepatocellular carcinoma (HCC) and intracellular cholangiocarcinoma (ICC) is essential for clinical management and prognostic prediction. However, non-invasive differential diagnosis between HCC and ICC remains highly challenging. Dynamic contrast-enhanced ultrasound (D-CEUS) with standardized software is a valuable tool in the diagnostic approach to focal liver lesions and could improve accuracy in the evaluation of tumor perfusion. Moreover, the measurement of tissue stiffness could add more information concerning tumoral environment. To explore the diagnostic performance of multiparametric ultrasound (MP-US) in differentiating ICC from HCC. Our secondary aim was to develop an US score for distinguishing ICC and HCC. Between January 2021 and September 2022 consecutive patients with histologically confirmed HCC and ICC were enrolled in this prospective monocentric study. A complete US evaluation including B mode, D-CEUS and shear wave elastography (SWE) was performed in all patients and the corresponding features were compared between the tumor entities. For better inter-individual comparability, the blood volume-related D-CEUS parameters were analyzed as a ratio between lesions and surrounding liver parenchyma. Univariate and multivariate regression analysis was performed to select the most useful independent variables for the differential diagnosis between HCC and ICC and to establish an US score for non-invasive diagnosis. Finally, the diagnostic performance of the score was evaluated by receiver operating characteristic (ROC) curve analysis. A total of 82 patients (mean age ± SD, 68 ± 11 years, 55 men) were enrolled, including 44 ICC and 38 HCC. No statistically significant differences in basal US features were found between HCC and ICC. Concerning D-CEUS, blood volume parameters (peak intensity, PE; area under the curve, AUC; and wash-in rate, WiR) showed significantly higher values in the HCC group, but PE was the only independent feature associated with HCC diagnosis at multivariate analysis (*p* = 0.02). The other two independent predictors of histological diagnosis were liver cirrhosis (*p* < 0.01) and SWE (*p* = 0.01). A score based on those variables was highly accurate for the differential diagnosis of primary liver tumors, with an area under the ROC curve of 0.836 and the optimal cut-off values of 0.81 and 0.20 to rule in or rule out ICC respectively. MP-US seems to be a useful tool for non-invasive discrimination between ICC and HCC and could prevent the need for liver biopsy at least in a subgroup of patients.

## 1. Introduction

Primary liver cancer remains a universal health burden as it is the sixth most common in prevalence and the fourth leading cause of cancer-related deaths worldwide [1,2]. Hepatocellular carcinoma (HCC) makes up approximately 80% of primary liver cancer, while intrahepatic cholangiocarcinoma (ICC) is the second most common subtype and its incidence has rapidly increased in recent years, accounting for 15–20% of all primary hepatic malignancies [3]. Although they share similar risk factors such as viral hepatitis and cirrhosis, the prognosis and long-term outcomes of patients with ICC are poorer than those with HCC [4]. Moreover, treatment options recommended for patients with HCC, such as local ablative therapy and liver transplantation, are not always suitable for patients with ICC [5,6]. Therefore, correct differentiation between HCC and ICC is essential for clinical management and prognostic prediction. However, difficulties in the differential diagnosis of ICC versus HCC present a long-disputed and controversial issue.

Diagnostic imaging modalities which assess contrast enhancement dynamics are essential for the characterization of incidentally detected liver lesions. Among them, contrast-enhanced ultrasound (CEUS) has been extensively applied since it allows a continuous view of the enhancement pattern of the lesions and the real-time evaluation of their perfusion [7]. It has been demonstrated that CEUS has excellent diagnostic accuracy for the non-invasive diagnosis of HCC in high-risk patients [8]. For this reason it has been included in current international guidelines for the diagnosis of HCC as a second-line imaging method when contrast-enhanced computed tomography (CT) or contrast-enhanced magnetic resonance imaging (MRI) do not allow the drawing of definitive conclusion or are contraindicated [1,9].

Many studies have investigated the diagnostic accuracy of CEUS in differentiating ICC from HCC and a recent meta-analysis summarized all current results, identifying six CEUS features which may be useful for this purpose [10]. Unfortunately, all ultrasound-based modalities rely on the experience of the reader and scanning conditions [11]. Moreover, one major issue is subjectivity in the assessment of enhancement patterns, which makes an accurate differential diagnosis of focal liver lesions difficult, even for experienced radiologists.

To standardize image interpretation and improve the accuracy of diagnosis of suspected HCC, the American College of Radiology released the contrast-enhanced ultrasound Liver Imaging Reporting and Data System (CEUS LI-RADS), which is now widely used as a diagnostic system for patients at risk of developing HCC [12]. Recently, its validity has also been demonstrated in patients without liver disease [13]. In this system, HCC should be distinguished not only from benign lesions but also from other malignant tumors. In particular, a category known as LI-RADS (LR) M hat includes malignant lesions without features typical of HCC (such as rim enhancement and early and/or marked wash-out) has been introduced. However, like CT/MRI, the CEUS LI-RADS cannot achieve optimal sensitivity for the non-invasive diagnosis of HCC [14]. This means that 25–40% of HCC patients cannot obtain a non-invasive diagnosis and require liver biopsy [15]. On the other hand, previous studies have demonstrated the high sensitivity (91.0–100.0%) but unsatisfying specificity (48.5–87.0%) of LR-M in establishing a diagnosis of ICC [13].

Therefore, attempts have been made to improve non-invasive diagnostic algorithms and objectivity in the assessment of contrast enhancement behavior. In particular, dynamic contrast-enhanced ultrasound (D-CEUS), a novel imaging method that allows quantitative evaluation of lesion microvasculature, has recently been introduced [16]. Several studies have shown high diagnostic accuracy, sensitivity and specificity of D-CEUS in the characterization of focal liver lesions [17,18,19,20]. Parameters reflecting arterial contrast enhancement such as high peak enhancement (PE) and short time to peak (TTP) have been described as characteristic of HCC [21]. On the other hand, a rapid contrast agent wash-out documented by a short mean transit time (mTT) is considered typical of ICC [22].

Although some reports have described the different radiological characteristics of primary liver tumors in contrast-enhanced imaging [23,24], the differential diagnosis between HCC and ICC remains highly challenging. 

It has been reported that combined examination by CEUS and shear wave elastography (SWE) in solid liver lesions might have a greater accuracy than CEUS alone [25].

Ultrasound elastography, integrated in conventional US systems and especially two-dimensional SWE (2D-SWE), which provides quantitative information about tissue stiffness, is a promising tool to differentiate malignant from benign focal liver lesions [26,27,28]. However, the elastic differences between different pathological types of malignant lesions are still controversial. For example, Park et al. [29] showed that ICC have low shear wave velocity, while other studies had the opposite results [30].

We hypothesize that a multiparametric US (MP-US) approach based on B-mode, elastography and D-CEUS might provide useful information for the non-invasive diagnosis of primary liver cancers. Hence, in the present study we aimed to investigate the relationship between different US features and histological diagnosis. In addition, we attempted to define an US-based score to improve the differential diagnostic performance between HCC and ICC.

## 2. Results

### 2.1. Patients Clinical Data

In accordance with the aforementioned inclusion and exclusion criteria, 85 consecutive patients with histological diagnoses proving HCC or ICC lesions gave written informed consent for the prospective evaluation and were screened for this study. Among them, 3 patients did not enter the study due to inadequate quality of CEUS clips. Therefore, 82 patients (55 men; 27 women; mean age ±SD: 68 ± 11 years) completed the study. Demographic and clinical data of the study population are provided in Table 1. There were no significant differences between HCC and ICC groups in terms of age, gender, tumor size or echogenicity on conventional ultrasound. Cirrhosis was more common in the HCC group (68.4% vs 27.3% in ICC lesions), whereas the serum level of CA19-9 was higher in the ICC patients (*p*< 0.01).

### 2.2. US Imaging Characteristics

Basal US features were not significantly different between HCC and ICC lesions (Table 1).

On CEUS, homogenous hyperenhancement in the arterial phase was observed in 68.5% (26/38) and 22.7% (10/44) of HCC and ICC lesions, respectively. Only one (2.6%) HCC nodule showed rim hyperenhancement versus 15 ICC nodules (34.1%) 

In total, 73 of 82 (89%) nodules showed wash-out in the portal or late phase. The mean (± SD) wash-out time was 59.4 ± 26.9 s in HCC compared to 45.4 ± 17.2 s in ICC lesions (*p* = 0.01). Marked wash-out was found in 28.9% (11/38) of HCC lesions, in contrast to 52.3% (23/44) of ICC lesions (*p* = 0.03).

Concerning D-CEUS, we found significant differences in the absolute values of four blood volume parameters: PE (*p* < 0.03), WiAUC, WoAUC and WiWoAUC (*p* < 0.01) (Table 2). When we considered the ratio between lesions and the surrounding parenchyma, PE and WiAUC remained significantly higher for HCC compared to ICC, but we also observed higher WiR values (*p* < 0.01). Conversely, mTTI, a parameter associated with portal and late-phase wash-out, was significantly shorter for ICC versus HCC (*p* = 0.03), whereas no significant differences were observed between RT and TTP in terms of tumor wash-in.

**Table 2 ijms-24-08548-t002:** Contrast-enhanced ultrasound qualitative and quantitative parameters stratified for type of neoplasia.

Parameter	Entire Cohort n = 82	HCC n = 38	ICC n = 44	*p* Value
**Arterial phase contrast enhancement**				**<0.01**
hyper- omogeneous	36 (43.9)	26 (68.5)	10 (22.7)	
aspecific	30 (36.6)	11 (28.9)	19 (43.2)	
rim	16 (19.5)	1 (2.6)	15 (34.1)	
**Wash-out (yes)**	73 (89.0)	31 (81.6)	42 (95.5)	**0.04**
marked wash-out (yes)	34 (41.5)	11 (28.9)	23 (52.3)	**0.03**
**Wash-out time (s)**	51.4 (22.8)	59.4 ± 26.9	45.4 ± 17.2	**0.01**
**Stiffness lesion (kPa)**	41.2 ± 15.1	35.7 ± 13.8	47.1 ± 14.4	**<0.01**
**PE lesion**	872 (314–1805)	1269 (636–2149)	487 (249–1751)	**0.03**
**PE Ratio**	0.27 (0.31–1.44)	0.66 (−0.08–1.74)	−0.04 (−0.53–1.33)	**<0.01**
**WiAUC lesion**	5802 (2670–12202)	9399 (5060–16057)	4145 (1493–9146)	**<0.01**
**WiAUC Ratio**	−0.35 (−0.67–−0.09)	−0.22 (−0.51–0.46)	−0.52 (−0.74–−0.26)	**0.01**
**RT lesion**	9.1 (7.1–12.7)	10.9 (7.9–13.0)	8.1 (6.9–12.4)	0.14
**mTTI lesion**	51.2 (26.9–98.7)	67.0 (47.8–118.0)	42.9 (32.2–94.4)	**0.04**
**TTP lesion**	11.4 (7.9–15.2)	12.5 (8.7–14.5)	9.1 (7.8–15.6)	0.27
**WiR lesion**	185 (63–522)	208 (77–625)	133 (62–333)	0.08
**WiR ratio**	1.75 (0.60–4.50)	2.41 (1.32–6.67)	1.32 (0.13–2.71)	**<0.01**
**WoAUC lesion**	19019 (5944–42896)	32048 (15406–62715)	12138 (3815–30984)	**<0.01**
**WoAUC Ratio**	−0.23 (−0.59–0.57)	−0.11 (−0.38–0.62)	−0.41 (−0.68–0.03)	0.08
**WiWoAUC lesion**	24894 (9085–55002)	42243 (20351–78362)	15779 (5489 −41443)	**<0.01**
**WiWoAUC Ratio**	−0.26 (−0.58–0.22)	−0.13 (−0.43–0.52)	−0.39 (−0.64–−0.11)	0.06

Median, 1 and 3 quartiles (non-normally distributed variables); mean ± standard deviation (normally distributed variables), number (%). Chi2, Student’s T-test or Mann–Whitney U test; PE (in Arbitrary Units, AU); area under the wash-in curve, WiAUC (in AU); wash-in rate, WiR (in AU); rise time, RT (in seconds); TTP (in seconds); mean transit time local, mTT (in seconds); area under the wash-out curve, WoAUC (in AU); area under the wash-in and wash-out curves, WiWoAUC (in AU).

**Table 3 ijms-24-08548-t003:** Univariate and multivariate logistic models predicting ICC.

	Univariate Model		Multivariate Model	
	OR (CI 95%)	*p* Value	OR (CI 95%)	*p* Value
**Gender (male)**	0.57 (0.22–1.46)	0.24		
**Age (years)**	0.97 (0.93–1.02)	0.25		
**BMI (Kg/m^2^)**	0.96 (0.87–1.05)	0.34		
**Liver cirrhsosis**	0.17 (0.07–0.45)	**<0.01**	0.17 (0.05–0.56)	**<0.01**
**Target nodule diameter (mm)**	1.02 (1.00–1.04)	**0.02**		
**aFP**	1.00 (0.99–1.01)	0.26		
**Ca 19.9**	1.00 (0.99–1.01)	0.26		
**Stiffness lesion (kPa)**	1.05 (1.02–1.09)	**<0.01**	1.05 (1.01–1.09)	**0.01**
**PE lesion**	1.00 (0.99–1.01)	0.13		
**PE Ratio**	0.62 (0.43–0.91)	**0.01**	0.54 (0.32–0.90)	**0.02**
**WiAUC lesion**	1.00 (0.99–1.01)	0.13		
**WiAUC Ratio**	0.73 (0.46–1.15)	0.18		
**RT lesion**	0.98 (0.91–1.06)	0.70		
**mTTI lesion**	1.00 (0.99–1.01)	0.87		
**TTP lesion**	1.01 (0.97–1.06)	0.55		
**WiR lesion**	0.99 (0.99–1.00)	0.06		
**WiR ratio**	0.87 (0.78–0.98)	**0.02**		
**WoAUC lesion**	1.00 (0.99–1.01)	0.12		
**WoAUC Ratio**	0.69 (0.42–1.12)	0.13		
**WiWoAUC lesion**	1.00 (0.99–1.01)	0.11		
**WiWoAUC Ratio**	0.65 (0.38–1.10)	0.11		

Median, 1 and 3 quartiles (non-normally distributed variables); mean ± standard deviation (normally distributed variables), number (%). Chi2, Student’s T-test or Mann–Whitney U test; alpha-fetoprotein (AFP); carbohydrate antgen19-9 (CA19-9); PE (in Arbitrary Units, AU); area under the wash-in curve, WiAUC (in AU); wash-in rate, WiR (in AU); rise time, RT (in seconds); TTP (in seconds); mean transit time local, mTT (in seconds); area under the wash-out curve, WoAUC (in AU); area under the wash-in and wash-out curves, WiWoAUC (in AU).

Finally, ICC had significantly higher stiffness values than HCC (47.1 ± 14.4 vs. 35.7 ± 13.8 kPa; *p* < 0.01).

Figure 1 and Figure 2 show the typical US features of HCC and ICC, respectively, with the corresponding time–intensity curves and stiffness values.

### 2.3. Diagnostic Performance of the US Score in Predicting ICC

According to the univariate analysis, liver cirrhosis (OR: 0.17, 95% CI: 0.07–0.45, *p* < 0.01), target nodule diameter (OR: 1.02, 95% CI: 1.00–1.04, *p* = 0.02), lesion stiffness (OR: 1.05, 95% CI: 1.02–1.09, *p* < 0.01), PE ratio (OR: 0.62, 95% CI: 0.43–0.91, *p* = 0.01) and WiR ratio (OR: 0.87, 95% CI: 0.78–0.98, *p* = 0.02) were all associated with histopathological diagnosis.

A multivariate logistic regression analysis was carried out to assess the influence of each parameter for the differential diagnosis between HCC and ICC. Three independent predictors were selected: liver cirrhosis (*p* < 0.01), SWE (*p* = 0.01) and PE ratio (*p* = 0.02) (Table 3). A score based on those variables was highly accurate for the differential diagnosis of primary liver tumors, with an area under the ROC curve of 0.836. Accordingly, the predictor equation for histopathological diagnosis would be: $$P=\frac{e^{x}}{\left(1+e^{x}\right)}$$
x = −1.7457 × cirrhosis + 0.0504 × SWE −0.6157 × PE ratio – 0.9168

Liver cirrhosis was present or not, with the values of 1 or 0, respectively, and SWE was expressed in kPa.

According to ROC analysis and the dual cut-off approach, the optimal cut-off values to rule in or rule out ICC were 0.81 and 0.20, respectively (Figure 3). This means that when US score < 0.20, the lesion would be diagnosed as HCC; otherwise, when US score > 0.81, the lesion would be diagnosed as ICC. 

There were 12 of 35 (34.3%) ICC patients and 17 of 37 (45.9%) HCC patients in the entire cohort correctly diagnosed by the US score, whereas a misclassification occurred only in 3 ICC (8.6%) and 3 HCC (8.2%) patients. 

## 3. Discussion

Focal liver lesions are a common finding in both cirrhotic and non-cirrhotic liver patients and the improvement of non-invasive diagnostic criteria is a common clinical concern for accurate diagnosis and treatment of liver cancer [31].

Using a second-generation contrast agent, CEUS is safe, cost-effective and performable immediately after recognition of a suspicious liver lesion by B-mode US [32,33]. Providing real-time visualization of contrast enhancement kinetics, it has the potential to evaluate tumor perfusion by assessing even weak intratumoral blood flow with high sensitivity.

Several guidelines for the management of patients at risk for the development of HCC allow imaging-based diagnosis without biopsy if typical imaging features are present in CEUS, dynamic CT or MRI [9,34]. However, the diagnostic accuracy of imaging techniques in the diagnosis of ICC is a controversial issue [35,36]. 

Moreover, current studies only focus on the analysis and improvement of imaging criteria for the diagnosis of HCC in high-risk patients [8]. Our study first combined clinical and multiparametic imaging data to explore and improve non-invasive diagnostic algorithm.

The combination of different imaging features together with clinical background could significantly improve the sensitivity of the diagnostic algorithm, thus potentially reducing the necessity of invasive biopsies in patients with liver nodules. Although our results were not perfect, the high AUC, sensitivity and specificity of the US score indicated promising discrimination ability and accuracy. 

In particular, we demonstrated that CEUS parameters combined with stiffness values and liver background may be useful for the differential diagnosis between HCC and ICC. 

It has been demonstrated that CEUS is able to improve the diagnostic accuracy of CT and MRI, allowing the characterization of nodules with atypical vascular patterns [37,38]. In particular, several studies focused on the identification of diagnostic features for ICC [31,32,33,34,35,36,37,38,39].

A recent meta-analysis performed to evaluate the diagnostic performance of CEUS showed high sensitivity (0.92) and specificity (0.87) in distinguishing between ICC and HCC with an elevated overall diagnostic ability (AUC 0.95) [10].

The typical CEUS findings of ICC are peripheral rim-like enhancement during the arterial phase and early and marked wash-out. However, the detection rate of peripheral rim-like enhancement of ICCs varies from 31.5 to 73.3% among different institutions [18,40]. In addition, some HCC also showed early wash-out on CEUS, thus complicating the diagnosis.

In recent years, the development of CEUS has attracted scientific interest as a method of more objective and possibly more sensitive assessment of focal liver lesions perfusion and characterization [40,41,42,43,44,45,46,47].

Because of the possibility of observing the flow of the contrast agent continuously, the exact maximum intensity at PE can be observed by D-CEUS in contrast to CT and MRI. Various quantitative perfusion parameters have been assessed for their value in differentiating and characterizing focal liver lesions. 

In a study by Wildner et al., time-related parameters of the portal venous and late phases showed significantly lower values in the ICC group, indicating early wash-out of the contrast agent, whereas PE was significantly higher in HCC lesions [46]. Our results confirmed the role of PE in the differential diagnosis between ICC and HCC but, in contrast with previous studies, the role of time-related parameters was less relevant for the diagnosis. 

Interestingly, in our study, several D-CEUS features were significantly different between HCC and ICC, although at multivariate analysis only PE was independently related to histological diagnosis. In order to reduce patient-related perfusion variability, we “normalized” blood volume-related D-CEUS parameters of the lesion to that of the surrounding liver parenchyma.

Moreover, this study extends the analysis of individual specific CEUS features to a predictive model-based differential diagnostic approach combining imaging and clinical data. 

Previous studies demonstrated that the contrast-enhanced pattern of primary liver tumors could be different in cirrhotic and non-cirrhotic livers in relation to changes in liver perfusion that occur in patients with chronic liver disease [48,49].

To date, there is little evidence in the literature concerning the relationship between liver perfusion and contrast enhancement kinetics. Few studies suggest that contrast enhancement in HCC and ICC in cirrhosis might differ from non-cirrhotic livers [48,49]. As expected, we found that liver cirrhosis was an independent risk factor for histological diagnosis and may be an important clue in distinguishing between HCC and ICC. 

Together with D-CEUS parameters and liver background also 2D-SWE was independently associated with lesion histology. As pathognomonically expected the stiffness of ICC was significantly higher than that of HCC in agreement with the results of previous studies [50,51,52]. We hypothesized that the combined examination by D-CEUS and 2D-SWE could increase the diagnostic accuracy of D-CEUS alone (*p* < 0.05). As a consequence, they were incorporated into a non-invasive predictive model easily applicable in clinical practice.

Our score, allowing the simultaneous evaluation of lesion perfusion, tumoral environment and clinical background could be more suitable than traditional imaging features for individualized precise predictions of primary liver tumors. Moreover, the application of a dual cut-off approach permits the maximization of sensitivity and specificity. In particular, we were able to correctly identify 34% of ICC and nearly 46% of HCC not classified with traditional imaging criteria. Of course, there are a subset of lesions, included in the gray zone of the score (between 0.20 and 0.81), that cannot be adequately diagnosed. In these cases, liver biopsy becomes necessary. However, if our results are confirmed in larger populations and in different cohorts, the score will allow non-invasive diagnosis in a subset of patients. 

### Limitations of the Study

Some limitations are associated with our study. First, we have not included all focal liver lesions. In fact, our specific aim was to improve the non-invasive diagnostic algorithm for primary liver cancers. However, it would be very interesting to test the feasibility of the US score in all types of liver lesions.

Second, this was a single-center study with a relatively small sample size and a heterogeneous collective (cirrhotic and non-cirrhotic patients), which may conceal a potential selection bias. Additional multicenter prospective studies are needed to validate the diagnostic criteria. Third, US is an operator-dependent technique with lower sensitivity in obese and meteoric patients; thus, all potential nodules may not be adequately evaluated with this non-invasive imaging modality. Moreover, there are technical limitations of D-CEUS. The decision to include as much tumor volume as possible inside the ROI could lead to more heterogeneity. Another issue is the individual situation of liver perfusion that limits the inter-individual comparability of D-CEUS values. In order to achieve greater comparability, the D-CEUS parameters of liver lesions were “normalized” by referring to liver parenchyma. 

Finally, the acquisition of D-CEUS clips and time–intensity curves as well as the whole analysis of parametric imaging are complicated and time-consuming; as a consequence, the feasibility and clinical usefulness of the method in clinical routine are still limited.

Despite of those limitations, MP-US seems to be a valuable tool for non-invasive diagnosis of primary liver tumors. 

## 4. Materials and Methods

### 4.1. Patients

Between January 2021 and September 2022, all consecutive patients with focal liver lesions referred to our interventional US unit for liver biopsy were evaluated for enrollment. 

Inclusion criteria were as follows: age older than 18 years, pathologic diagnosis of ICC or HCC confirmed by liver biopsy, liver lesion visible on conventional US. 

Patients with extrahepatic cholangiocarcinoma, patients with combined HCC-cholangiocarcinoma and patients who underwent therapy before US evaluation were excluded from the study. Other exclusion criteria were hypersensitivity to Sonovue, no pathological classification or poor image quality. 

The protocol was approved by our Institutional Review Board (protocol number 2824). All patients provided written informed consent.

### 4.2. Study Protocol

This is an uncontrolled, prospective, monocentric study including patients with focal liver lesions at an Italian university hospital serving as a tertiary referral center for the diagnosis and treatment of liver diseases.

For the purpose of the study, patients were evaluated on the same day with a liver biopsy with standard US, D-CEUS and 2D-SWE. 

Basic clinical data, including age and sex, were recorded. Laboratory tests included hepatitis status, alpha-fetoprotein (AFP) levels and carcinoembrionic antigen 19-9 (CA 19-9) levels. 

Liver cirrhosis was diagnosed by histopathological analysis or imaging methods [53]. 

The primary goal was the assessment of differences in US parameters and their correlation with pathological diagnosis. The secondary goal was the development of an US-based multiparametric score for non-invasive imaging diagnosis.

### 4.3. Multiparametric US Examination

US studies were performed by an experienced physician (M.E.A. with 13 years of experience in liver imaging) with an Aixplorer Mach 30 (SuperSonic Imagine, Aix-en-Provence, France) equipped with a wideband C6-1 convex probe (frequency range, 1.0–6.0 MHz). 

After a fasting status of at least 6 h, US examination was performed in three phases. First, the morphologic study was performed in B-mode US in order to identify the lesion and evaluate basal US features. In cases where more than one lesion was present in a single patient, only the lesion with the best US visualization was chosen. 

Subsequently, 2D-SWE was performed with the same equipment. Measurements were obtained with the patient in a supine position and suspended normal breathing. The shear wave measurement box was positioned on the liver lesion in order to obtain an appropriate SWE map. In the case of very large lesions, we placed the region of interest (ROI) in the most representative part of the focal lesion. Margins and central necrosis in larger lesions were spared. Elasticity values are displayed in a color-coded 2D image of tissue stiffness in box form over a conventional B-mode image. Only the cases with a stability index (SI) of at least 90% were considered reliable. This is a quality parameter derived from spatial and temporal stiffness stability within the circular Q-Box. The median values of at least 3 successful measurements were considered for analysis and was expressed in kilopascals (kPa). An interquartile range to the median ratio (IQR/M) < 30% was used as a measurement reliability criterion.

After B-mode and SWE analysis, a functional study was conducted and a single bolus of 2.4 mL of sulfur hexafluoride-filled microbubbles (SonoVue, Bracco, Milan, Italy) was injected as an intravenous bolus, followed by a flush of 10 mL normal saline, using a 20-gauge venous catheter that had been inserted into the cubital vein at the level of the left antecubital fossa. A dedicated, contrast-specific, continuous-scanning, low-mechanical index technique (MI = 0.08) was used in order to study the whole vascular phase consisting of the arterial (0–30 s), portal (31–120 s) and late (until 5 min) phases according to EFSUMB guidelines [54]. The imaging timer was started immediately upon completion of SonoVue injection. Overall gain was set to obtain a complete anechoic image of the lesion for the basal phase, depth was regulated according to the patient’s habitus and focus position was always set below the area to be examined. In order to avoid motion artefacts, patients were asked to perform relaxed breathing during video clip acquisition. Signal enhancement of the lesion was evaluated in real time and a dynamic sequence of 3 min was recorded continuously on a hard disk and exported as Digital Imaging and Communications in Medicine (DICOM) image for further analysis. 

### 4.4. Image Analysis

All US images, including B-mode US, SWE and CEUS, were reviewed by two operators who had more than 10 years of experience in US liver imaging (MAZ and MEA). Neither patient details nor clinical or pathological results were available to the physicians. 

The following imaging features were used to categorize each nodule: size, echogenicity, boundary, pattern of arterial phase enhancement, presence, timing and degree of wash-out. The physicians read the image features independently, and disagreements were resolved by discussion and consensus. 

Compared to the enhancement levels of surrounding liver parenchyma, the wash-in patterns of target lesions during the arterial phase were classified as follows: homogeneous hyperenhancement, heterogeneous hyperenhancement, peripheral rim-like hyperenhancement, isoenhancement, and hypoenhancement [39].

The wash-out was classified into mild and marked wash-out according to its degree. Wash-out time was calculated from the time the contrast first appeared to the time when the nodule began to show hypo-enhancement compared with the surrounding hepatic parenchyma.

Finally, digitized quantification of contrast uptake was performed on the recorded video clip using the quantitative analysis software package VueBox, Version 7.4 (Bracco, Italy). The analysis can display the mean, median and the standard deviation of intensity pixel within the ROI drawn on the image for each frame of the sequence acquired. In our study, the time–intensity curves were generated from a manually defined ROI placed over the lesion and large enough to encompass as much of the lesion volume as possible. A second ROI, serving as a potential internal reference, was drawn on the adjacent liver tissue at the same depth of the lesion.

Respiratory movement artifacts were eliminated by automatic adjustments and, when necessary, by deleting selected frames in the post-processing analysis. Quantification was performed on uncompressed linear data (raw data), which is linearly proportional to microbubble concentration. The results were expressed as intensity values after calculating the arithmetic mean of pixel intensities. A gamma variate fit was used as a statistical model to normalize the dispersion of gamma values in the perfusion analysis. Further details regarding D-CEUS and quantification processes are reported in a position paper endorsed by the European Federation of Societies for Ultrasound in Medicine and Biology (EFSUMB) [16].

Eight perfusion parameters were selected to characterize both blood volume and blood flow and were extracted from time–intensity curves: PE (in Arbitrary Units, AU); area under the wash-in curve, WiAUC (in AU); wash-in rate, WiR (in AU); rise time, RT (in seconds); TTP (in seconds); mean transit time local, mTT (in seconds); area under the wash-out curve, WoAUC (in AU); and area under the wash-in and wash-out curve, WiWoAUC (in AU). 

In order to reduce intra- and interindividual perfusion variability, blood volume-related parameters (PE, WiR and AUC) were expressed as a ratio considering the perfusion of the surrounding liver parenchyma as a reference. 

The entire quantification process, from drawing ROI to extrapolation of functional parameters, was performed by the two authors (MAZ and MEA) and the mean values of the two measurements were considered for analysis.

### 4.5. Reference Standard

Histopathological examination was the reference standard of this study. The bioptical samples were fixed in 10% neutral buffered formalin and embedded in paraffin. The tissue slices were stained with hematoxylin-eosin. Immunohistochemical staining for cytokeratin (CK) 19 and CK7 were used to identify ICC, whereas AFP and glypican 3 were used to identify HCC. Pathologic evaluations were performed by a pathologist with more than ten years of experience with liver cancer. 

### 4.6. Statistical Analyses

Statistical analysis was performed with STATA^®^ (Version 14.1, Stata Corporation; College Station, TX, USA). Descriptive statistics were presented as absolute numbers and percentages for discrete variables, median with interquartile range (IQR) for non-normally distributed continuous variables and mean ± standard deviation (SD for normally distributed continuous variables. Association between clinical, laboratory and US variables with type of neoplasia were assessed with the chi-squared test, Student’s T test or the Mann–Whitney test when appropriate. 

For blood volume-related CEUS quantitative parameters (PE, AUC and WiR), the ratio between lesion and parenchyma values was also calculated [(lesion—parenchyma)/parenchyma]. Unadjusted logistic regression analysis was performed for each predictive variable to assess the association with ICC diagnosis. Odds ratio (OR) estimates for the selected variables were reported together with 95% confidence intervals and Wald test *p* values.

The selection of parameters for ICC diagnosis probability score construction was based on a multivariable logistic regression model. Clinical, laboratory and US parameters were combined into a multivariable logistic regression model with a forward stepwise selection procedure to select the optimal parameters. Only factors associated with the univariate logistic regression model (*p* value threshold 0.1) were included in the multivariable logistic regression model. The model with the lowest Aikake information criterion (AIC) and Bayesian information criterion (BIC) was selected. Ten patients were not included in model construction because of missing data.

Performance of the score was assessed by the goodness of fit and discrimination ability. The goodness of fit (the agreement between observed outcome and prediction) was evaluated using calibration plot and Hosmer–Lemeshow test. Discrimination ability was assessed using receiver operating characteristic (ROC) curve analysis and measuring the area under the curve (AUC). Cut-offs proposed were obtained by applying the maximum Youden Index criterion and the dual cut-off approach. For the dual cut-off approach, optimal rule out (high sensitivity, >90%) and rule in (high specificity, >90%) cut-offs were selected. When evaluating performance at a given cut-off, sensitivity, specificity, positive predictive value (PPV) and negative predictive value (NPV) were computed.

## 5. Conclusions

D-CEUS, through allowing an accurate observation of perfusion characteristics, can display significant differences between malignant liver tumors. This is possible particularly by means of PE. The development of an US score including lesion stiffness and liver background further improves the diagnostic efficiency of D-CEUS. However, the differentiation between lesion entities, especially between ICC and HCC, remains challenging in a group of patients and, in cases of strong doubt, liver biopsy remains the reference method. 

Further studies are warranted to validate these results and to evaluate the performance of this US score in different populations and clinical setting.

## Figures and Tables

**Figure 1 ijms-24-08548-f001:**
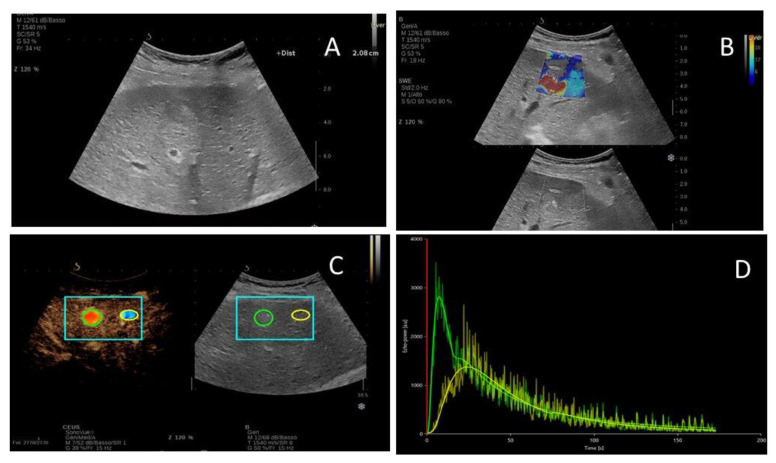
Ultrasound features of a 73 years old patient with HCC nodule of the segment VI. (**A**) Hypoechoic nodule of 2 cm diameter in B-mode ultrasound; (**B**) two-dimensional shear wave elastography image of the liver lesion; (**C**,**D**) contrast-enhanced ultrasound with corresponding time–intensity curves of the lesion (green) and liver parenchyma (yellow).

**Figure 2 ijms-24-08548-f002:**
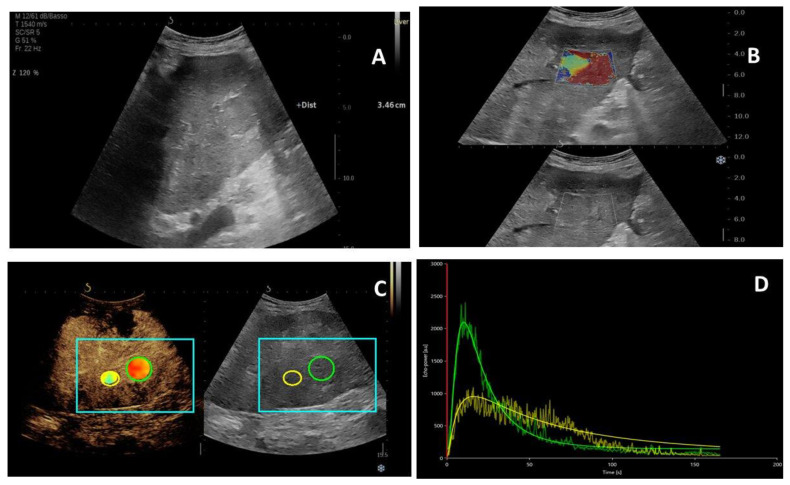
Ultrasound features of a 65 years old patient with ICC nodule of the segment IV. (**A**) Hypoechoic nodule of 3.5 cm diameter in B-mode ultrasound; (**B**) two-dimensional shear wave elastography image of the liver lesion; (**C**,**D**) contrast-enhanced ultrasound with corresponding time–intensity curves of the lesion (green) and liver parenchyma (yellow).

**Figure 3 ijms-24-08548-f003:**
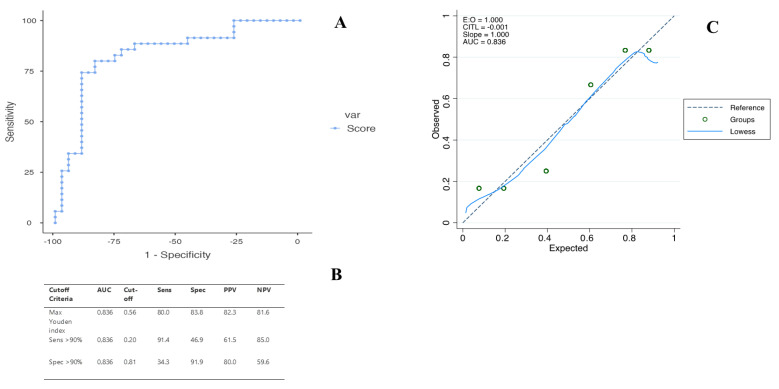
(**A**) Receiver operating characteristic curves of the ultrasound score for the differential diagnosis of HCC and ICC. (**B**) Cut-off values with corresponding sensitivity, specificity, PPV and NPV for diagnosis of ICC. (**C**) The calibration plot showing agreement between observed proportion (Y axis) and predicted probabilities (X axis). Circles represent participants grouped by similar predicted risk (six equal groups). The dotted line represents the ideal calibration. The solid line is the calibration estimated on the data using locally estimated scatterplot smoothing (LOESS). *Hosmer–Lemeshow goodness of fit p = 0.42*.

**Table 1 ijms-24-08548-t001:** Demographic, clinical and ultrasonographic characteristics of the entire cohort and stratified for type of neoplasia.

Parameter	Entire Cohort *n* = 82	HCC *N* = 38	ICC *N* = 44	*p* Value
**Gender (male)**	55 (67.1)	28 (73.7)	27 (61.4)	0.23
**Age (years)**	68 ± 11	70 ± 10	67 ± 11	0.25
**BMI (Kg/m^2^)**	25.9 ± 4.9	26.5 ± 4.8	25.4 ± 5.0	0.34
**Liver disease**				**<0.01**
normal liver	33 (40.2)	6 (15.8)	27 (61.4)	
chronic hepatitis	4 (4.9)	2 (5.3)	2 (4.5)	
steatosis	7 (8.5)	4 (10.5)	3 (6.8)	
cirrhosis	38 (46.3)	26 (68.4)	12 (27.3)	
**Nodules**				0.18
single	42 (51.2)	23 (60.5)	19 (43.2)	
2	12 (14.6)	4 (10.5)	8 (18.2)	
3	6 (7.3)	4 (10.5)	2 (4.5)	
>3	22 (26.8)	7 (18.4)	15 (34.1)	
**Target nodule diameter (mm)**	40 (30–60)	25 (23–45)	54 (35–74)	
**Echogenicity**				0.66
hypo-	58 (70.7)	26 (68.4)	32 (72.7)	
hyper-	18 (22.0)	10 (26.3)	8 (18.2)	
iso-	1 (1.2)	0 (0)	1 (2.3)	
mixed	5 (6.1)	2 (5.3)	3 (6.8)	
**Nodule margins**				0.06
regular	61 (74.4)	32 (84.2)	29 (65.9)	
irregular	21 (25.6)	6 (15.8)	15 (34.1)	
**aFP**	4.78 (2.22–51.0)	8 (3.25–85)	3.7 (2–13.5)	0.09
**Ca 19.9**	84.0 (12.1–576.0)	8 (4–32)	141 (33–3621)	**<0.01**

Median, 1 and 3 quartiles (non-normally distributed variables); mean ± standard deviation (normally distributed variables), number (%). Chi2, Student’s T-test or Mann–Whitney U test; alpha-fetoprotein (AFP); carbohydrate antgen19-9 (CA19-9).

## Data Availability

The data presented in this study are available on request from the corresponding author.

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
