# Peer review of "Multiparametric Dynamic Ultrasound Approach for Differential Diagnosis of Primary Liver Tumors"

_ijms, 2023, doi:10.3390/ijms24108548_

Round 1

Reviewer 1 Report

Dear authors,

The manuscript you have proposed for publication is interesting, but it still needs some improvement to be accepted for publication.

Minor :

- please revise the abstract; it has multiple mistakes

-the manuscript requires an English revision, and some paragraphs should be rephrased as they are very hard to follow, especially the Discussion section

-a conclusion section is lacking

Author Response

Please revise the abstract; it has multiple mistakes

Re: According to the reviewer observation we revised the abstract. We apologize for the blameworthy lack of attention during the written of the first draft.

The manuscript requires an English revision, and some paragraphs should be rephrased as they are very hard to follow, especially the Discussion section

Re: According to reviewer suggestion, we have revised the text with special attention to English language by one of the authors (Dr. Garcovich) who is a native English/German speaking expert.

A conclusion section is lacking

Re: Following the suggestion of the reviewer, we added the “Conclusion” section.

Reviewer 2 Report

This is a single-center prospective study investigating the efficacy of dynamic contrast-enhanced ultrasound (CE-US) for differential diagnosis in liver tumor (hepatocellular carcinoma vs. intrahepatic cholangiocarcinoma). The authors concluded that multiparametric dynamic US approach showed better diagnostic power compared to conventional B-mode US. However, the following concerns need to be addressed:

Major and minor comments

1.         The gold standard in diagnosing liver tumor is not US but dynamic CT/MRI. As the authors wrote in Introduction, US is a second-line imaging modality for liver tumor used for patients with inconclusive diagnosis in CT/MRI. There might be a benefit of low cost and no X-Ray exposure in CE-US, however the usefulness of CE-US in diagnosing liver tumor compared to dynamic CT/MRI need to be shown.

2.         The authors excluded patients with poor imaging in US from this study, however it is not appropriate. Doesn’t it show the inferiority of US compared to CT/MRI in diagnosing liver tumor?

3.         The patients with combined HCC/ICC also should be included in this study.

4.         The authors’ opinion was based on only univariate analysis.

5.         The clinical implication of this study is a little bit unclear. If the dynamic CE-US approach is useful in diagnosing liver tumor, could the authors omit tumor biopsy?

Author Response

The gold standard in diagnosing liver tumor is not US but dynamic CT/MRI. As the authors wrote in Introduction, US is a second-line imaging modality for liver tumor used for patients with inconclusive diagnosis in CT/MRI. There might be a benefit of low cost and no X-Ray exposure in CE-US, however the usefulness of CE-US in diagnosing liver tumor compared to dynamic CT/MRI need to be shown.

Re: The reviewer raises an important question as in the diagnostic algorithm of focal liver lesions CT/MRI are performed before liver biopsy and, in some cases, they are sufficient for conclusive diagnosis. However, we selected only patients with inconclusive imaging diagnosis who were scheduled for liver biopsy. This is why we didn’t compare US to CT/MRI but directly to histology.

The authors excluded patients with poor imaging in US from this study, however it is not appropriate. Doesn’t it show the inferiority of US compared to CT/MRI in diagnosing liver tumor?

Re: We agree with the reviewer that CT/MRI are the gold standard for detailed imaging of the liver and had greater sensitivity than US. However, in our series, the percentage of inadequate quality of US images is very low (< 4%) and does not affect the reliability of the results. Of course, the exclusions of some patients may introduce a selection bias and we have discussed this aspect as a limitation of the study.

The patients with combined HCC/ICC also should be included in this study.

Re: The suggestion of the reviewer to include patients with combined HCC/ICC is very tempting. Unfortunately among our study population only one patient had combined tumor.  Therefore it is not possible to perform statistical analysis or to draw solid conclusions from these unbalanced subgroups of patients.  Further studies on larger cases will permit to better clarify the role of the US score also in patients with combined HCC/ICC

The authors’ opinion was based on only univariate analysis.

Re: Following the suggestion of the reviewer we focused the discussion on parameters independently associated to histological diagnosis at multivariate analysis.

The clinical implication of this study is a little bit unclear. If the dynamic CE-US approach is useful in diagnosing liver tumor, could the authors omit tumor biopsy?

Re: We thanks the reviewer for raising this point. As we have additional pointed out in the paper, the US score could allow non invasive diagnosis in a subgroup of patients. However, if the results will be validated in further studies, the chance to avoid liver biopsy, at least in some patients, would be of great clinical value.

Round 2

Reviewer 1 Report

The authors adressed all my comments. The manuscript has improved a lot. Congratulations!

Author Response

According to reviewer suggestion, we have revised the text with special attention to English language